# Persistent activity in a recurrent circuit underlies courtship memory in *Drosophila*

Xiaoliang Zhao[1], Daniela Lenek[2], Ugur Dag[1], Barry J Dickson[1,3], Krystyna Keleman[1,2]*

[1]Janelia Research Campus, Ashburn, United States; [2]Research Institute of Molecular Pathology, Vienna, Austria; [3]Queensland Brain Institute, University of Queensland, St Lucia, Australia

**Abstract** Recurrent connections are thought to be a common feature of the neural circuits that encode memories, but how memories are laid down in such circuits is not fully understood. Here we present evidence that courtship memory in *Drosophila* relies on the recurrent circuit between mushroom body gamma (MBγ), M6 output, and aSP13 dopaminergic neurons. We demonstrate persistent neuronal activity of aSP13 neurons and show that it transiently potentiates synaptic transmission from MBγ>M6 neurons. M6 neurons in turn provide input to aSP13 neurons, prolonging potentiation of MBγ>M6 synapses over time periods that match short-term memory. These data support a model in which persistent aSP13 activity within a recurrent circuit lays the foundation for a short-term memory.

DOI: https://doi.org/10.7554/eLife.31425.001

*For correspondence:
kelemank@janelia.hhmi.org

**Competing interests:** The authors declare that no competing interests exist.

## Introduction

As animals pursue their goals, their behavioral decisions are shaped by memories that encompass a wide range of time scales: from fleeting working memories relevant to the task at hand, to short-term and long-term memories of contingencies learned hours, days, or even years in the past. Working memory is thought to reflect persistent activity generated within neural networks, including recurrent circuits (*Wang, 2001*). In contrast, short-term memory (STM) and long-term memory (LTM) involves changes in synaptic efficacy due to functional and structural modification of synaptic connections (*Kandel, 2001*). However, the neural circuit mechanisms involved in the formation, persistence and transitions between these distinct forms of memory are not fully known.

A robust form of memory in *Drosophila* is courtship memory, which can last from minutes to days, depending on the duration and intensity of training (*Siegel and Hall, 1979*; *McBride et al., 1999*). Naïve *Drosophila* males eagerly court both virgin females, which are generally receptive, and mated females, which are not (*Manning, 1967*; *Wolfner, 2003*). However, upon rejection by mated females, they become subsequently less likely to court other mated females (*Tompkins, 1984*). This selective suppression of courtship towards mated females, called courtship conditioning, can be attributed to the enhanced sensitivity of experienced males to an inhibitory male pheromone deposited on the female during mating, cis-vaccenyl acetate (cVA) (*Keleman et al., 2012*).

Olfactory memory in insects relies on the function of a central brain structure called the mushroom body (MB) (*de Belle and Heisenberg, 1994*; *Heisenberg et al., 1985*). The principal MB cells, the cholinergic Kenyon cells (KCs) (*Barnstedt et al., 2016*), receive input from sensory pathways in the dendritic calyx region and from dopaminergic neurons (DANs) in the axonal lobes of the MB. These MB lobes are compartmentalized, with each compartment innervated by specific classes of DANs and MB output neurons (MBONs) (*Aso et al., 2014a*; *Mao and Davis, 2009*). MBONs receive input from both KCs and DANs (*Takemura et al., 2017*).

**eLife digest** Memories help to shape behavior, and can last from a few seconds to an entire lifetime. Working memory, in which information is temporarily held available for use in an ongoing task, is the most fleeting form of memory. It relies on persistent activation of a network of nerve cells or neurons that represent the information in question. Strengthening the connections between those neurons may result in a longer-lasting memory. But the mechanisms that support the formation of memories of different durations are not fully understood.

Zhao et al. have now explored these mechanisms in the fruit fly by studying memory for courtship behavior. Inexperienced male fruit flies will attempt to court both virgin females and females who have recently mated. But the latter reject courtship attempts, and male fruit flies therefore learn to avoid them. This is known as courtship memory, and it relies on a network of neurons within a region of the fruit fly brain called the mushroom body.

Within the mushroom body, dopamine neuron sends signals to a neuron called the Kenyon cell, which in turn sends signals to a mushroom body output neuron. The latter activates circuits responsible for decision-making and movement. But it also activates the dopamine neuron, thereby forming a recurrent circuit or loop. When the courtship is rejected, the dopamine neuron becomes persistently active, which generates a working memory of the experience. If the circuit is activated again during this period of persistent firing, the working memory may be converted into a longer-lasting memory.

The results of Zhao et al. provide insights into the mechanisms by which memories form and undergo strengthening. They suggest that distinct processes within a single neural circuit give rise to memories of different durations. Recurrent loops are also present within the brains of mammals. Similar processes may thus support the formation and persistence of our own memories.

DOI: https://doi.org/10.7554/eLife.31425.002

We previously established that short-term courtship conditioning is mediated by the aSP13 class of DANs (also known as the PAM-γ5 neurons, [*Aso et al., 2014a*]), which innervate the MBγ5 compartment. The activity of aSP13 neurons is essential for courtship conditioning in experienced males and sufficient to induce conditioning in naïve males (*Keleman et al., 2012*). Here we demonstrate that courtship memory also requires the corresponding MBγ KCs and the MBγ5 MBONs, the glutamatergic M6 neurons (also known as MBON-γ5β'2a neurons [*Aso et al., 2014a*]). Furthermore, we present evidence that MBγ, M6, and aSP13 neurons form a recurrent circuit and that persistent activity of the aSP13 neurons mediates plasticity at the MBγ to M6 synapses that can last from minutes to hours. Consistent with this model, M6 activity is required not only for memory readout but also, like aSP13, for memory formation. These data support a model in which persistent aSP13 activity within the MBγ>M6>aSP13 recurrent circuit lays the foundation for short-term courtship memory.

## Results

### Courtship experience modulates circuit properties between MBγ and M6 neurons

We confirmed the involvement of MBγ and M6 neurons in courtship conditioning by chronically silencing them using cell-type specific GAL4 drivers (*Figure 1—figure supplement 1*) to express tetanus toxin light chain (UAS-TNT, an inhibitor of synaptic transmission; [*Martin et al., 2002*]). Single males of each genotype were trained by first pairing them for 1 hr with a single mated female, and then testing their courtship towards a fresh mated female after a 30 min rest period. We used automated video analysis to derive a courtship index (CI) for each male, defined as the percentage of time over a 10 min test period during which the male courts the female. A suppression index (SI) was then calculated as the relative reduction in the mean courtship indices of trained ($CI^+$) versus naïve ($CI^-$) populations: $SI = 100*(1-CI^+/CI^-)$. Control flies expressing an inactive form of tetanus toxin (UAS-TNTQ) typically showed a SI of ~40–50% (*Figure 1A,B*; *Supplementary file 1*). By contrast, males in which M6 neurons or MBγ neurons were silenced with an inhibitory form of tetanus toxin (UAS-TNT) showed much less or no suppression (*Figure 1A,B*; *Supplementary file 1*).

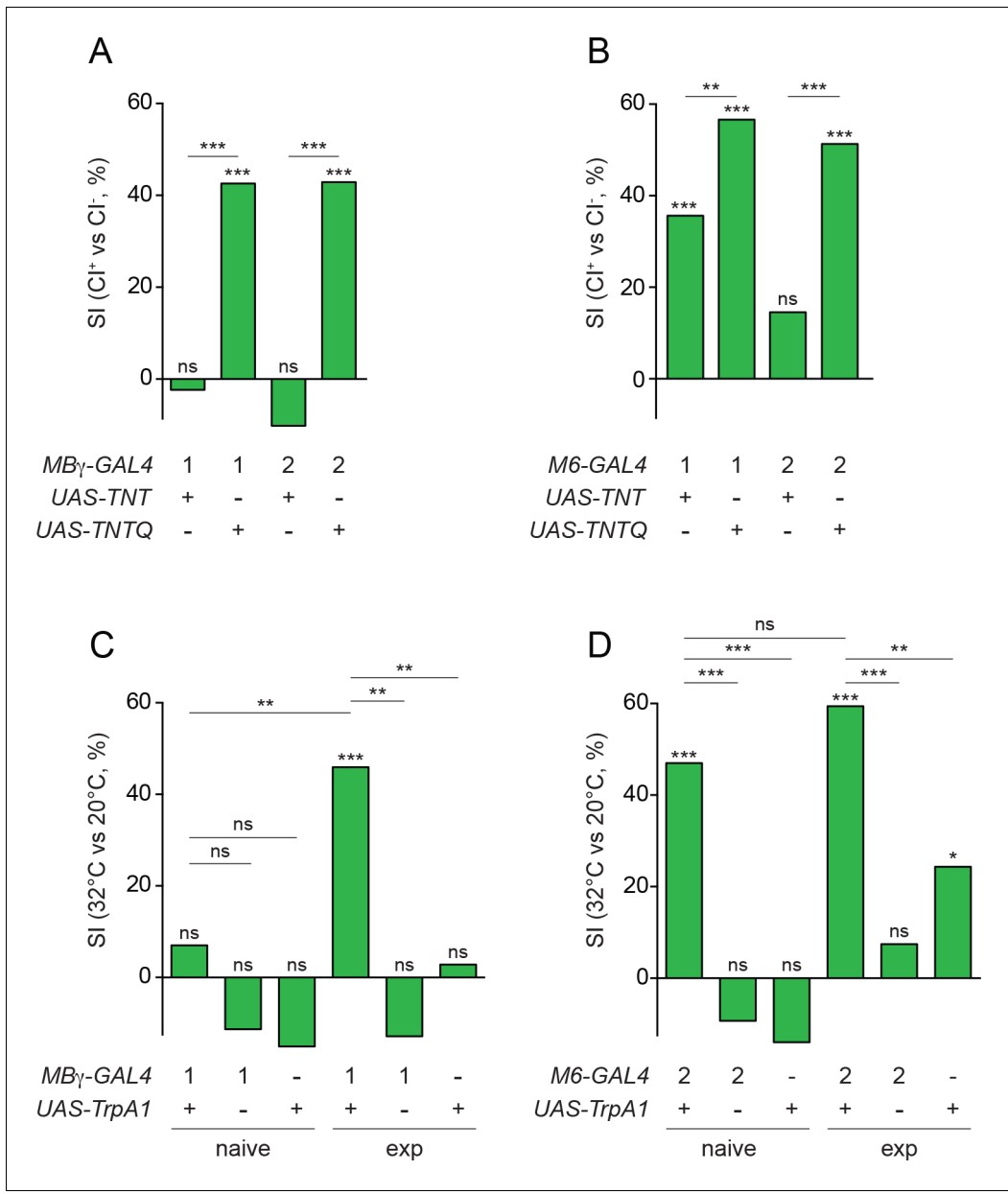

**Figure 1.** Experience modulates circuit properties between MBγ and M6 neurons. (**A**) Suppression indices (SI), calculated from mean courtship indices of male flies in which active (*UAS-TNT*) or inactive (*UAS-TNTQ*) tetanus toxin is expressed in MBγ neurons (1, *VT044966-GAL4*; 2, *VT030413-GAL4*). In this and other panels, statistical significance of differences from zero or from control groups is indicated as follows: ***p<0.001, **p<0.01, *p<0.05, n.s. p>0.05, permutation tests, see *Supplementary file 1*. (**B**) Suppression indices (SI), calculated from mean courtship indices of male flies in which active (*UAS-TNT*) or inactive (*UAS-TNTQ*) tetanus toxin is expressed in M6 neurons (1, *VT014702-GAL4*; 2, *VT032411-GAL4*). See *Supplementary file 1*. (**C**) Suppression indices (SI) of naïve or experienced (exp) male flies upon thermogenetic activation of MBγ neurons (1, *VT044966-GAL4*). See *Supplementary file 1*. (**D**) Suppression indices (SI) of naïve or experienced (exp) male flies upon thermogenetic activation of M6 neurons (2, *VT032411-GAL4*). See *Supplementary file 1*.
DOI: https://doi.org/10.7554/eLife.31425.003

The following figure supplement is available for figure 1:

**Figure supplement 1.** MBγ, M6 and aSP13 GAL4 driver lines.
DOI: https://doi.org/10.7554/eLife.31425.004

The DAN inputs to a given MB compartment are believed to modulate synaptic transmission from MB neurons to MBONs, primarily through their presynaptic inputs onto the KCs (*Kim et al., 2007*; *Qin et al., 2012*). Some studies have indicated that DANs enhance KC>MBON transmission (*Cohn et al., 2015*; *Owald et al., 2015*; *Plaçais et al., 2013*; *Pai et al., 2013*), whereas others have suggested that DANs depress these synapses (*Aso et al., 2014a*; *Hige et al., 2015*; *Owald et al., 2015*; *Séjourné et al., 2011*; *Hattori et al., 2017*; *Lewis et al., 2015*). The sign of modulation may therefore depend upon the context. We predicted that, if M6 is the relevant MBON for courtship conditioning, then artificial activation of M6 should suppress courtship. Moreover, if MBγ>M6 transmission is modified by training, then M6 activation should be equally potent in experienced and naïve males, whereas MBγ activation should be either more or less potent in experienced males, depending upon whether training potentiates or depresses MBγ>M6 synapses.

We tested these predictions using the thermosensitive cation channel TrpA1 (open at 32°C and closed at 20°C) (*Rosenzweig et al., 2005*) to activate either MBγ or M6 cells. To measure the extent of courtship suppression we used unreceptive virgin females (pseudomated females) as testers, which do not elicit significant courtship suppression in experienced males (*Keleman et al., 2012*). For each condition, we determined a SI as the percentage reduction in courtship activity towards these unreceptive virgins in 10 min assays performed at 32°C compared to 20°C: $SI = 100*(1-CI^{32}/CI^{20})$. We found that MBγ activation was significantly more potent in experienced males than in naïve males, in which it had only a small effect on courtship (*Figure 1C*; *Supplementary file 1*). By contrast, M6 activation suppressed male courtship with equal potency in both naïve and experienced males (*Figure 1D*; *Supplementary file 1*). We conclude from these data that courtship experience with mated females potentiates synaptic transmission from MBγ to M6 cells.

## Dopamine modulates synaptic transmission from MBγ to M6 neurons

To examine synaptic transmission between MBγ and M6 neurons, we used optogenetics. We generated a step-function channelrhodopsin variant, SFOCatCh, that combines mutations to increase the off kinetics (SFO = C128S/D156A, [*Yizhar et al., 2011*]) with a single amino acid substitution to enhance the conductance of divalent cations (CatCh = L132C, [*Kleinlogel et al., 2011*]). We validated SFOCatCh by whole-cell patch clamp recording in olfactory projection neurons (*Figure 2—figure supplement 1*). We used SFOCatCh in conjunction with GCaMP6s (*Chen et al., 2013*) to monitor calcium responses in whole explanted brains of naïve males. The combination of SFOCatCh and GCaMP6s temporally uncouples the optical inputs required for activity manipulation and calcium imaging. In all experiments with SFOCatCh and GCaMP6s reported here, we imaged calcium responses during three consecutive 4 s periods, each of which was preceded by a 100 ms pulse of green, blue, or green light, respectively, to turn SFOCatCh OFF, ON, or OFF again (*Figure 2A*). This protocol thus provides a pre-stimulus baseline, a during-stimulus response, and a post-stimulus response. To assess whether and how dopamine modulates MBγ>M6 transmission, we repeated this OFF/ON/OFF protocol 3 times at 3 min intervals: first without dopamine, then with either 0.1 mM or 1 mM dopamine delivered for the first second of each imaging period through a perfusion pipette positioned at the γ5 compartment, and finally following dopamine washout (*Figure 2A*).

We could not detect any robust calcium response in either the dendrites (*Figure 2B–D*) or axon termini (*Figure 2—figure supplement 2*) of M6 when we activated MBγ with SFOCatCh in the absence of exogenous dopamine. However, a strong dose-dependent calcium response was consistently observed in trials with dopamine during the SFOCatCh ON period. In contrast, little or no response was observed during either the SFOCatCh OFF periods (*Figure 2B–D*, *Figure 2—figure supplement 2*) or the SFOCatCh ON period after dopamine washout (*Figure 2B,C*, *Figure 2—figure supplement 2*). We obtained similar results when we applied the dopamine receptor agonist apomorphine rather than dopamine (*Figure 2E*), or used CsChrimson (*Klapoetke et al., 2014*) rather than SFOCatCh as the optogenetic activator (*Figure 2—figure supplement 3*). The response to dopamine and SFOCatCh was completely abolished by application of the nicotinic acetylcholine receptor antagonist mecamylamine (*Figure 2F*), which blocks synaptic transmission from KCs to MBONs (*Barnstedt et al., 2016*). Together, these data indicate that cholinergic synaptic transmission from MBγ to M6 cells is initially weak but can be acutely potentiated by dopamine.

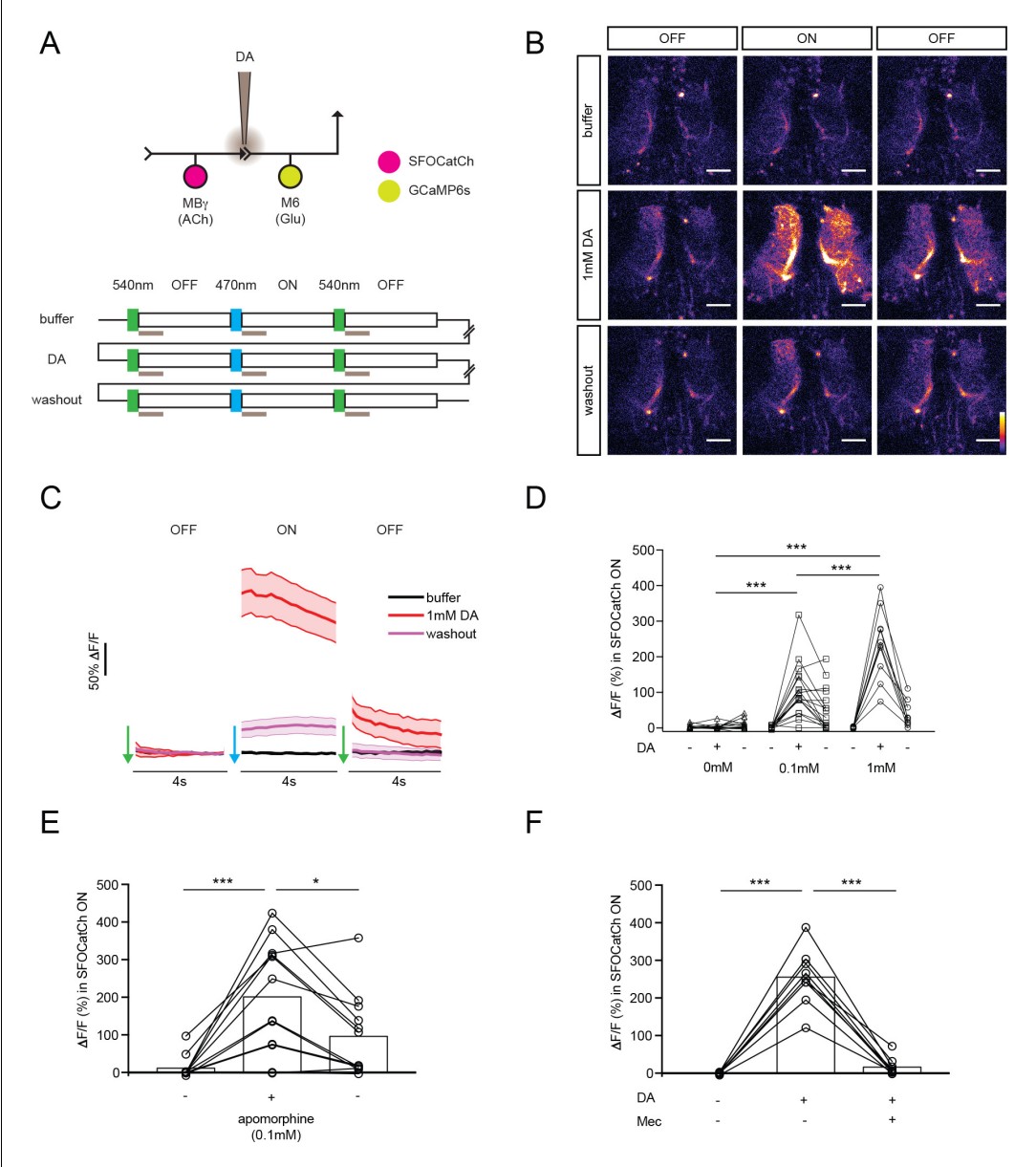

Figure 2. Dopamine modulates synaptic transmission from MBγ to M6 neurons. (A) Experimental protocol. OFF and ON indicate 4 s imaging periods, preceded by 100 ms pulses of 540 nm or 470 nm light to switch SFOCatCh OFF and ON, respectively. Gray bars indicate 1 s focal perfusion into the γ5 compartment. Buffer, dopamine injection (DA) and washout trials are separated by 3 min intervals. (B) Representative calcium responses in M6 dendrites in the γ5 compartment. Scale bar, 10 μm. (C) Average ΔF/F responses in M6 dendrites. n = 10. Mean ± s.e.m. (D) Average ΔF/F responses during the SFOCatCh ON periods of successive buffer, DA, and washout trials. n = 15, 17, 10 for 0, 0.1, and 1.0 mM DA, respectively. ***p<0.001, t-test. (E) Average ΔF/F responses during the SFOCatCh ON periods of successive buffer, apomorphine, and washout trials. n = 12. *p<0.05, ***p<0.001, t-test. (F) Average ΔF/F responses during the SFOCatCh ON periods of successive trials with buffer only, 1 mM DA, and DA plus 0.15 mM mecamylamine (Mec). n = 9. ***p<0.001, t-test.

DOI: https://doi.org/10.7554/eLife.31425.005

The following figure supplements are available for figure 2:

Figure supplement 1. SFOCatCh, a step-function optogenetic activator.
DOI: https://doi.org/10.7554/eLife.31425.006
Figure supplement 2. Calcium responses in M6 axons upon optogenetic stimulation of MBγ neurons using SFOCatCh.
DOI: https://doi.org/10.7554/eLife.31425.007
Figure supplement 3. Calcium responses in M6 neurons upon optogenetic stimulation of MBγ neurons using CsChrimson.
DOI: https://doi.org/10.7554/eLife.31425.008

## Repetitive stimulation of MBγ potentiates MBγ to M6 transmission

Whereas we could not detect a strong calcium response in M6 MBONs upon MBγ activation in the absence of dopamine, others have observed calcium responses in various MBONs, including M6, upon activation of KCs without application of dopamine or DAN stimulation (*Cohn et al., 2015*; *Owald et al., 2015*). We noted however that in our initial control experiments without dopamine, in which we sometimes performed multiple trials on the same sample, a calcium response could indeed be detected in the later trials. This suggests that stimulus history may account for some of the variability in MBON responses to KC stimulation in the absence of dopamine or DAN activation. To explore this possibility more rigorously, we activated MBγ neurons with SFOCatCh using the same OFF/ON/OFF protocol as before, now repeating the stimulus every minute. The initial stimuli, as previously observed in the trials without exogenous dopamine, did not elicit a detectable GCaMP6s response in M6 neurons. However, after 3–4 trials a significant calcium response was observed (*Figure 3A,B*). This response increased upon each successive stimulation before reaching a plateau after approximately 20 trials. This response was blocked by the dopamine D1-type receptor antagonist SCH23390 (*Figure 3C*), regardless of whether it was applied during the induction or plateau phase. This suggests that, upon repetitive stimulation of MBγ neurons, endogenous dopamine enables synaptic transmission to M6 neurons. The most likely source of this endogenous dopamine supply is the aSP13 neurons. Indeed, by shifting GCaMP6s from M6 to aSP13, we confirmed that the aSP13 DANs respond in a similar manner as M6 to the repetitive activation of MBγ neurons (*Figure 3E,F*)

To determine how long MBγ>M6 synapses remain potentiated after repetitive MBγ activation, we first induced potentiation with 30 pulses of MBγ activation at 1 min intervals, and then examined the response of M6 neurons to a single pulse of MBγ activation after 1, 2 or 3 hr. Potentiation at MBγ>M6 synapses was barely diminished after 1 hr, but fell to about 50% of its initial level after 3 hr (*Figure 3D*). The persistence of potentiation at MBγ>M6 synapses in these experiments is thus in line with the persistence of the courtship memory after a 30 min training period (*Keleman et al., 2012*).

## Activation of the MBγ>M6>aSP13 recurrent circuit elicits persistent aSP13 activity

Anatomically, DANs and MBONs innervating the same MB compartment, have the potential to form recurrent loops, with MBONs providing input to DANs (*Aso et al., 2014a*; *Takemura et al., 2017*; *Ichinose et al., 2015*; *Eichler et al., 2017*; *Owald et al., 2015*). In particular, the axonal termini of M6 MBONs are closely apposed to the aSP13 dendrites (*Aso et al., 2014a*). We therefore tested whether activation of M6 neurons elicits a calcium response in aSP13 neurons by expressing SFO-CatCh in M6 and GCaMP6s in aSP13. Indeed, acute activation of M6 neurons produced a strong calcium response in aSP13 (*Figure 4A,B*). This response was blocked by the NMDA receptor antagonist AP-5 (*Figure 4C*), consistent with glutamatergic transmission from M6 cells. Activation of MBγ neurons with SFOCatCh also elicited a strong calcium response in aSP13 neurons (*Figure 4E,F*) that was also dependent on glutamatergic neurotransmission, as well as both cholinergic transmission and dopamine (*Figure 4G*).

Whereas the M6 response to MBγ activation was diminished in the post-stimulus OFF period in trials with dopamine (*Figure 2B,C*), the response of aSP13 neurons to either M6 or MBγ activation persisted into the post-stimulus SFOCatCh OFF period (*Figure 4A,B,E and F*). In each case, the calcium response in aSP13 gradually declined over a 2 min period (*Figure 4D and H*). The persistent response of aSP13 neurons is not an intrinsic property of aSP13 neurons, since it was not observed when SFOCatCh was used to activate the aSP13 neurons themselves (*Figure 4—figure supplement 1*). Given that the response of aSP13 to MBγ or M6 activation is blocked by AP-5, we infer that this persistent activity is induced by glutamatergic transmission from M6 cells.

The persistent release of dopamine by aSP13 neurons for several minutes after stimulation could create a time window during which MBγ to M6 transmission is facilitated. Activation of MBγ neurons during this time window, as in our repetitive SFOCatCh activation experiments, would thus lead to further activation of M6 and aSP13, thereby creating a recurrent feedback circuit. We lack a reliable tripartite genetic means to test directly whether silencing aSP13 neurons blocks the GCaMP6 response in M6 upon repetitive activation of SFOCatCh in MBγ. We could confirm, however, that

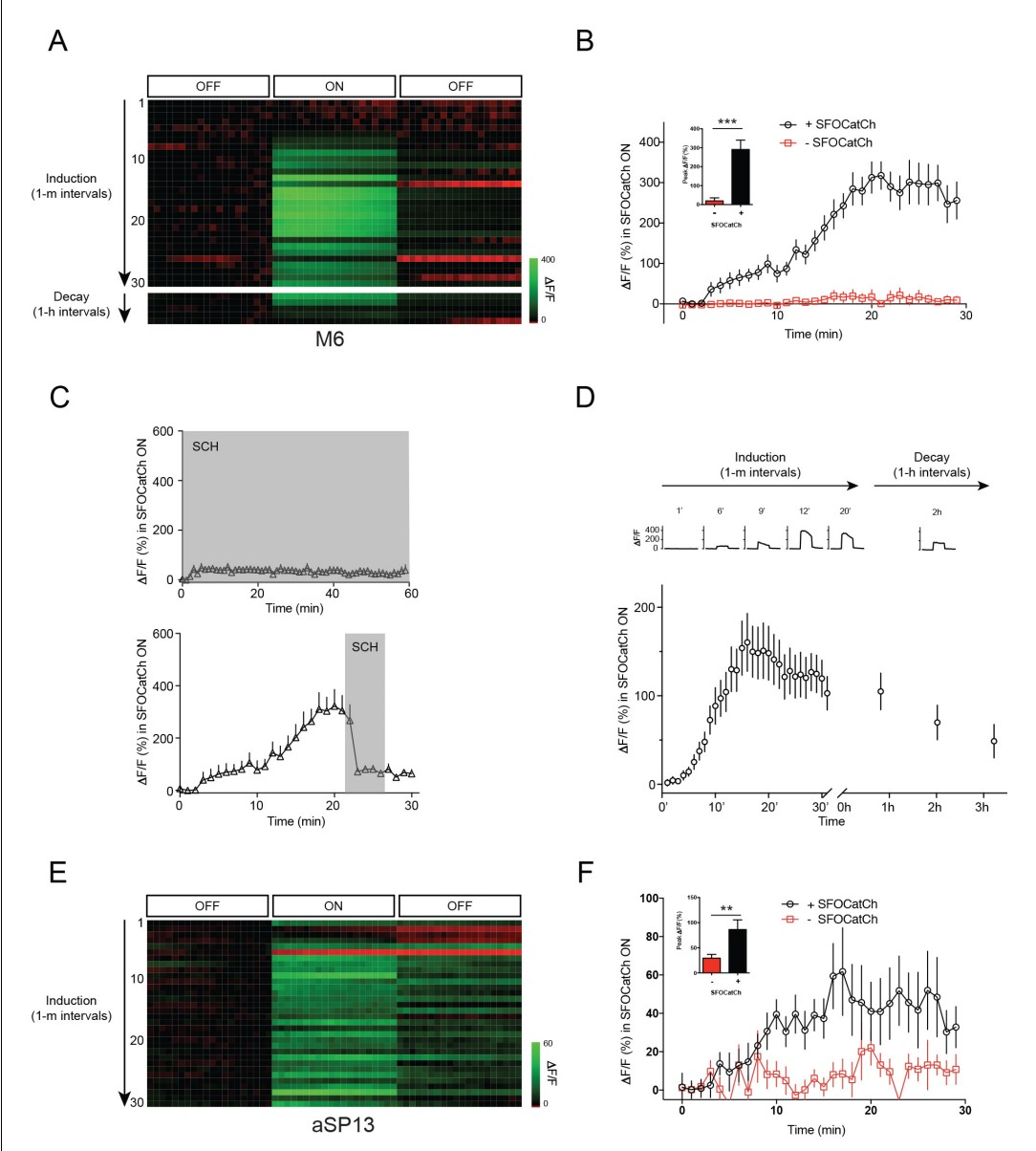

**Figure 3.** Repetitive stimulation of MBγ potentiates MBγ to M6 transmission. (**A**) Representative calcium responses in M6 dendrites in the γ5 compartment upon repetitive optogenetic stimulation of MBγ neurons with SFOCatCh. (**B**) Time course of average ΔF/F responses in M6 dendrites during potentiation, mean ± s.e.m. *n* = 16 for +SFOCatCh, *n* = 6 for –SFOCatCh. Inset, peak ΔF/F responses, mean ± s.e.m. ***p<0.001, t-test. (**C**) Average ΔF/F responses during the SFOCatCh ON periods in M6 dendrites in the γ5 compartment in trials with SCH23390 present (grey shading) either during (top) or after (bottom) induction. (**D**) Time course of average ΔF/F responses in M6 dendrites during potentiation (30 stimuli at 1 m intervals) and decay (stimulation at ~1 hr intervals), mean ± s.e.m. *n* = 19. Top, representative calcium responses at various time points, during the 3 × 4 s OFF/ON/ OFF imaging periods. (**E**) Representative calcium responses in aSP13 axons in the γ5 compartment upon repetitive optogenetic stimulation of MBγ neurons with SFOCatCh. (**F**) Time course of average ΔF/F responses in aSP13 axons during potentiation, mean ± s.e.m. *n* = 10 for +SFOCatCh, *n* = 7 for –SFOCatCh. Inset, peak ΔF/F responses, mean ± s.e.m. **p<0.01, t-test.

DOI: https://doi.org/10.7554/eLife.31425.009

this prolonged M6 response is blocked by AP-5 (*Figure 4—figure supplement 2*), which inhibits NMDA-type glutamate receptors and the persistent response of aSP13 (*Figure 4C*).

Activity of aSP13 neurons is strictly required during the training period of courtship conditioning (*Keleman et al., 2012*). The data presented here suggest that activation of aSP13 during training could open a time window of a several minutes during which MBγ>M6 transmission is facilitated. In

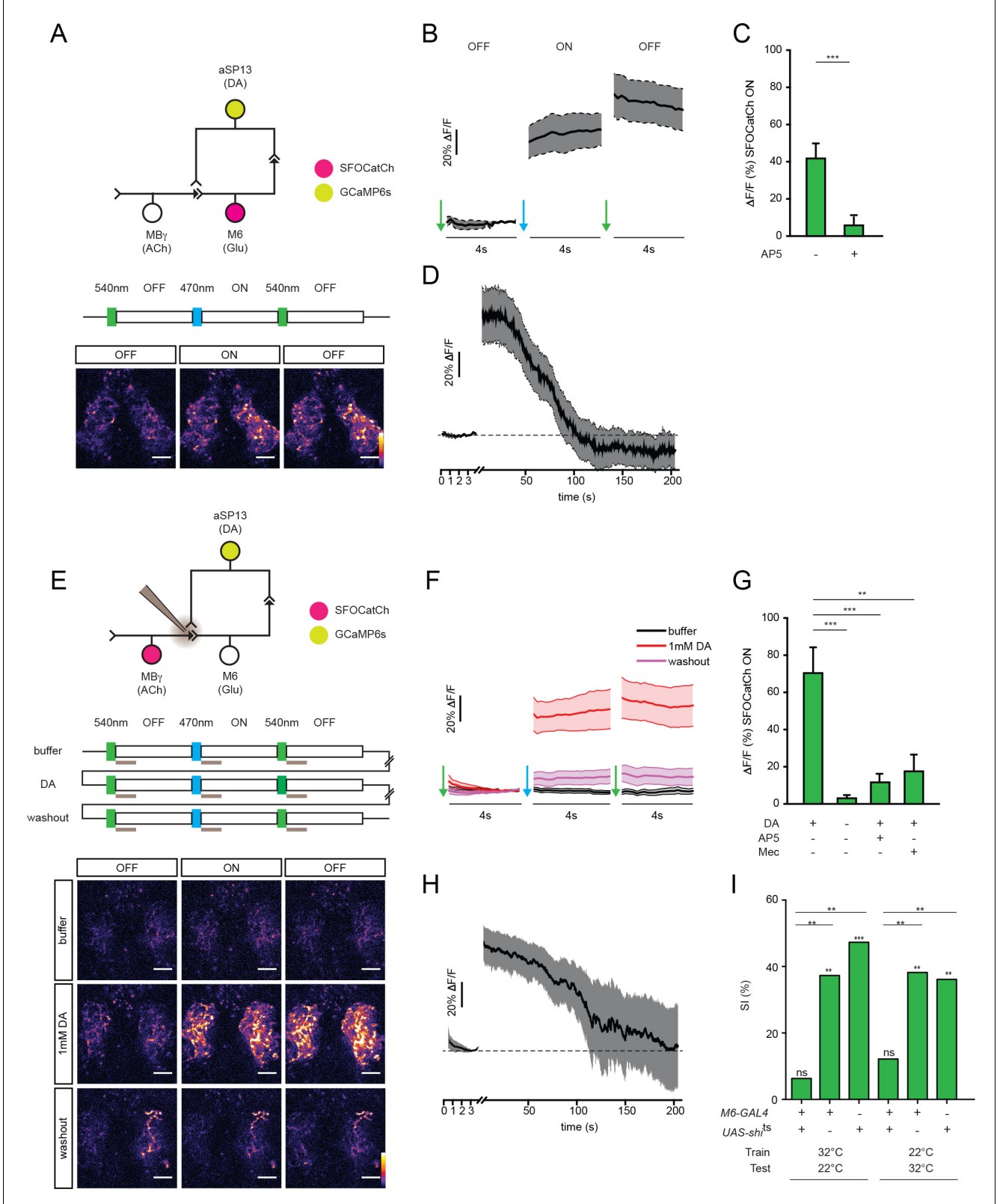

**Figure 4.** M6 or MBγ activation induces a persistent calcium response in aSP13. (**A**) Experimental protocol for M6 activation and aSP13 imaging, and representative calcium responses in aSP13 axons in the γ5 compartment. Scale bar, 10 μm. (**B**) Average ΔF/F responses in aSP13 axons, mean ± s.e.m. $n$ = 22. (**C**) Average ΔF/F responses during the SFOCatCh ON periods in trials with ($n$ = 9) or without 50 μM D-AP-5 ($n$ = 22). ***p<0.001, t-test. (**D**) Average ΔF/F responses, imaged at 1 Hz after 200 s of post-stimulus section. (**E**) Experimental protocol for MBγ activation and aSP13 imaging, and

*Figure 4 continued on next page*

Figure 4 continued

representative calcium responses in aSP13 axons in the γ5 compartment. Scale bar, 10 μm. (F) Average ΔF/F responses in aSP13 axons, mean ± s.e.m. n = 13. (G) Average ΔF/F responses during the SFOCatCh ON periods in trials with or without 1 mM DA, 50 μM AP-5, or 150 μM mecamylamine (Mec). n = 13, 13, 5,10, respectively. **p<0.01, ***p<0.001, t-test. (H) Average ΔF/F responses, imaged at 1 Hz after 200 s of post-stimulus section. (I) Suppression indices (SI) of male flies in which shi$^{ts}$ is expressed in M6 neurons, shifted to 32°C during training or testing, as indicated. ***p<0.001, **p<0.01, *p<0.05, permutation tests, see **Supplementary file 2**.

DOI: https://doi.org/10.7554/eLife.31425.010

The following figure supplements are available for figure 4:

**Figure supplement 1.** Stimulation of aSP13 does not elicit a persistent autonomous calcium response.
DOI: https://doi.org/10.7554/eLife.31425.011
**Figure supplement 2.** Response in M6 after repetitive activation of MBγ is blocked by NMDA-R antagonist.
DOI: https://doi.org/10.7554/eLife.31425.012

our training paradigm, males usually court mated females in a series of brief courtship on and off periods that could repetitively activate MBγ in the time window when aSP13 neurons are persistently activated and thus engage the recurrent MBγ>M6>aSP13 circuit, thereby potentiating MBγ>M6 transmission for a period of 2–3 hr. This leads to the somewhat counterintuitive prediction that M6 should not only act in memory retrieval, as generally assumed for MBONs, but should also be required for memory formation. To test this prediction, we conditionally silenced M6 neurons with a temperature-sensitive inhibitory form of dynamin (*UAS-shi$^{ts}$*), which blocks synaptic transmission at 32°C but not at 22°C (*Kitamoto, 2002*). Single males were trained and tested as before, but kept at 22°C except for a brief shift to 32°C either during training or during testing. Males in which M6 neurotransmission was blocked either during training or during testing had suppression indices indistinguishable from 0. Thus, whereas synaptic transmission from aSP13 is only required during memory acquisition (*Keleman et al., 2012*), M6 output is required during both acquisition and recall (*Figure 4I*; *Supplementary file 2*). We therefore propose that STM formation requires the MBγ>M6>aSP13 recurrent circuit, whereas readout occurs through other M6-dependent pathways.

## Discussion

In this study we have identified and characterized a tripartite MBγ>M6>aSP13 recurrent circuit that is essential for courtship memory in *Drosophila*. Our behavioral and physiological data suggest the following model for the function of this feedback loop in short-term courtship memory. When a naïve male courts a mated female, the aSP13 and MBγ neurons may both be activated, perhaps in response to behavioral rejection and olfactory stimuli presented by the female, respectively. Dopamine released by aSP13 neurons potentiates transmission from MBγ to M6 neurons, which in turn provide a recurrent excitatory glutamatergic input back onto aSP13 neurons. Upon activation by M6, aSP13 activity persists for several minutes, providing a short time window during which continued MBγ activity can further drive M6 and aSP13. Thus sustained, aSP13 activity can lead to a longer-lasting accumulation of dopamine in the γ5 compartment, facilitating MBγ>M6 neurotransmission for up to 2–3 hr.

The timescales for these physiological processes in ex vivo brain preparations broadly match the dynamics of courtship training and short-term memory formation. In our standard training paradigm, the male typically courts the female over several minutes, during which he performs a series of courtship bouts, each lasting for several seconds. As a result, a behavioral memory forms that lasts for several hours (*Keleman et al., 2012*). Memory formation during training requires both M6 and aSP13, consistent with the notion that it reflects activation of the recurrent circuit (*Figure 4* and [*Keleman et al., 2012*]). Memory readout requires M6 but not aSP13 (*Figure 4* and [*Keleman et al., 2012*]), and so evidently does not involve the recurrent circuit. We infer that M6 suppresses courtship through other, aSP13-independent, pathways, and that its ability to do so is independent of experience. The consequence of training is to provide MBγ neurons with access to this M6-dependent courtship suppression pathway (*Figure 1*).

Two important open questions are, first, what mechanism underlies the persistent calcium response in aSP13, and second, how does potentiation of MBγ>M6 synapses result in enhanced sensitivity to cVA, the hallmark of courtship memory (*Keleman et al., 2012*). The persistent response in

aSP13 is evidently not an intrinsic property of aSP13, as it is not induced when aSP13 neurons themselves are activated. This observation would also likely exclude reciprocal excitation between aSP13 and other DANs (*Plaçais et al., 2012*). Persistent aSP13 activity is induced in response to transient M6 activation, and is not associated with any persistent activity of M6 neurons themselves. Thus, it is also unlikely to involve feedback from aSP13 and M6, although aSP13 >M6 synapses likely do exist (*Eichler et al., 2017*; *Lin et al., 2007*). One possibility is that aSP13 persistence reflects unusually prolonged activation of the glutamatergic M6 >aSP13 synapses, or perhaps lies within interposed but still unidentified circuit elements.

Given that M6 neurons activate a courtship suppression pathway, the potentiation of MBγ>M6 neurotransmission may explain why MBγ activation suppresses courtship in trained but not naïve flies. But MBγ neurons likely do not specifically respond to cVA (*Caron et al., 2013*; *Gruntman and Turner, 2013*), so this change alone cannot account for the enhanced sensitivity of trained flies to cVA. A small and variable subset of MB γneurons do receive input from the olfactory pathway that processes cVA, but cVA is not required during training (*Keleman et al., 2012*) and it is difficult to envision any other mechanism by which aSP13-dependent plasticity could be specifically restricted to the cVA-responsive MBγ neurons. It is formally possible that, despite the broad potentiation of MBγ output synapses upon training, it is only the contribution of the cVA-responsive MBγ neurons that drives courtship suppression when the male subsequently encounters as mated female. Alternatively, it has been suggested that M6 neurons encode a generic aversive signal (*Aso et al., 2014b*), and so specificity to cVA might instead arise in downstream circuits that selectively integrate M6 output with the innate cVA-processing pathway from the lateral horn. In this regard, it is interesting to note that other MBONs have been implicated in courtship learning (*Montague and Baker, 2016*) or general aversion (*Aso et al., 2014b*), but M6 is the only MBON common to both.

Late activation of the same aSP13 neurons in the time window of 8–10 hr after training is both necessary and sufficient to consolidate STM to LTM (*Krüttner et al., 2015*). Thus, in the time window when STM would otherwise decay (*Keleman et al., 2007*), reactivation of the same MBγ>M6>aSP13 recurrent circuit may instead consolidate it into LTM. The mechanism by which aSP13 neurons are reactivated is unknown, but is evidently dependent upon their activation within the MBγ>M6>aSP13 recurrent circuit during training. It will be interesting to find out how this late aSP13 reactivation mechanism might relate to the mechanism that underlies persistent aSP13 activity during training.

In summary, our data suggest that a brief persistent activity of aSP13 neurons represents a neural correlate of courtship working memory, while the prolonged potentiation of MBγ>M6 synapses corresponds to STM. We propose that persistent activity of the dopaminergic neurons in the MBγ>M6>aSP13 feedback loop lays the foundation for formation of short-term courtship memory in *Drosophila,* and that later reactivation of the same recurrent circuit consolidates STM into LTM. Thus, in contrast to the prevailing view of memory progression in the Drosophila MB that distinct memory phases are located in different compartments or lobes (*Aso and Rubin, 2016*; *Davis, 2011*; *Pascual and Préat, 2001*), our data suggest that in the context of courtship conditioning, working memory, STM, and LTM all reside in the same γ5 compartment. Our conclusions do not preclude however, the involvement of other MB neurons in courtship memory (*Montague and Baker, 2016*) as it is conceivable that modulation, potentially of the opposite sign, of the appetitive memory pathways could be critical for courtship learning (*Perisse et al., 2016*). We therefore envision that distinct courtship memory types are not located in distinct circuits, but rather mediated by distinct processes within a common circuit. Encoding distinct memory phases within a common circuit may be an efficient mechanism for encoding memories for which the behavioral consequence is largely independent of timing and context (*Fusi et al., 2005*).

## Materials and methods

### Fly strains

Flies for behavior experiments were reared in vials with standard cornmeal food at 25°C, or as indicated, at 60% humidity in a 12 hr:12 hr light:dark cycle. Flies for physiological experiments were reared on standard cornmeal food, supplemented with 500 µM all-trans-retinal, in dark.

For behavioral and physiological experiments we used VT-Gal4 and VT-LexA lines obtained from the VT library, a collection of 2 kb enhancer fragments, generated following the strategy of (*Pfeiffer et al., 2008*) (B.J.D., unpublished data), *UAS-Kir2.1* (*Nitabach et al., 2002*), *UAS-TNT/UAS-TNTQ* (*Martin et al., 2002*), *UAS-Shi*^ts (*Kitamoto, 2002*), *UAS-TrpA1* (*Rosenzweig et al., 2005*), *UAS-SFOCatCh* (VIE-260b) (B.J.D., unpublished), 20x*UAS-CsChrimson-tdTomato* (*SuHwattp5*) and *LexAop2-opGCaMPs* (*SuHwattp1*) (gift from Barret Pfeiffer), *LexAop*-IVS-GCaMP6s-p10 (*attp1*) (*Chen et al., 2013*). Pseudomated females were (*elav*-Gal4/+*UAS*-SP/+) virgins (*Keleman et al., 2012*).

## Behavior

Courtship conditioning was performed as described (*Siwicki and Ladewski, 2003*). For training, solitary males (aged for 5–6 days) were placed in food chambers for 1 hr either with (trained) or without (naïve) a single mated female. After training each male was recovered, allowed to rest for 30 min and tested with a fresh mated female. Tests were performed in 10 mm diameter chambers and videotaped for 10 min (JVC handyman, 30 GB HD). We used automated video analysis to derive a courtship index (CI) for each male, defined as the percentage of time over a 10 min test period during which the male courts the female.

## Statistics

A MATLAB script (permutation test) (*Kamyshev et al., 1999*) was used to for statistical comparison of SIs between two groups. Briefly, the entire set of courtship indices for both naïve and trained flies were pooled and then randomly assorted into simulated naïve and trained groups of the same size as the original data. A SI was calculated for each of 100,000 randomly permutated data sets, and P values were estimated for the null hypothesis that learning equals 0 ($H_0$: SI = 0) or for the null hypothesis that experimental and control males learn equally well ($H_0$: SI = $SI_c$).

## Immunohistochemistry

Fly brains and ventral nerve cords were dissected in Schneider's insect medium and fixed in 2% paraformaldehyde (PFA) at room temperature for 55 min. Tissues were washed in PBT (0.5% Triton X-100 in phosphate buffered saline (PBS)) and blocked using 5% normal goat serum) before incubation with antibodies (diluted in blocking solution in a volume of 200 µl per sample). Primary antibodies (rabbit anti-GFP A-11122 from Molecular Probes at 2 µg/ml and mouse anti-Bruchpilot nc82 hybridoma supernatant from DSHB at 1 µg/ml) were applied for 2–3 days. After a rinse and four 15 min washes in PBT, tissues were then incubated for 2–3 days with Alexa Fluor 488-conjugated goat anti-rabbit and Alexa Fluor 568-conjugated goat anti-mouse secondary antibodies (Molecular Probes; 2.5 µg/ml and 5 µg/ml, respectively). Each of the antibody incubations were done for 4 hr at room temperature before placing the samples at 4 ˚C for the remainder of the incubation time. After a rinse and four 15 min washes in PBT, tissues were fixed with 4% PFA in PBS for 4 hr, followed by a rinse and four 15 min washes in PBT. Directly before mounting, tissues were rinsed and washed for 15 min in PBS to remove the Triton. The tissues were mounted on poly-L-lysine-coated cover slips and then dehydrated with 10 min ethanol baths of 30%, 50%, 75%, 95% and 3 × 100% followed by three 5 min washes in 100% xylene. Finally, mounted samples were embedded in xylene-based mounting medium (DPX; Electron Microscopy Science, Hatfield, PA) and dried for 2 days. Images were collected using an LSM710 confocal microscope (Zeiss, Germany) fitted with a Plan-Apochromat 20x/0.8 M27 objective.

## SFOCatCh

SFOCatCh was constructed from a synthetic ChR2 open reading frame with codon usage optimized for *Drosophila*, using mutagenic PCR to introduce the C128S and D156A substitutions to make it switchable (*Yizhar et al., 2011*) and the L132C mutation to increase cation conductance (*Kleinlogel et al., 2011*). The resulting coding fragment was inserted into a modified UAS vector for site-specific insertion at the VIE-260b landing site.

## Two-photon calcium imaging

For ex vivo calcium imaging experiments, 5–7 days old naïve males were briefly anesthetized on ice and brains were dissected out in calcium free external saline (ES) containing: 103 mM NaCl, 3 mM KCl, 5 mM TES (N-tris[hydroxymethyl]methyl-2-aminoethane sulfonic acid, a buffer chemical with peak performance around pH7.5), 10 mM trehalose, 10 mM glucose, 26 mM $NaHCO_3$, 1 mM $NaH_2PO_4$, 4 mM $MgCl_2$, 7 mM sucrose, pH 7.4, 275 mOsm (*Gu and O'Dowd, 2006*). The brain explants were transferred into a custom-made imaging chamber and mounted with anterior side up. Brains were perfused with ES supplemented freshly with 2 mM calcium, at speed 2 mL/min, pre-saturated with mixture of 95% $O_2$/5% $CO_2$. All two-photon imaging were performed using 40x N.A. 0.75 water-immersion objective (N-Achroplan, Zeiss), on LSM 7 MP microscope (Zeiss) with a Ti:sapphire laser (Chameleon Vision II, Coherent, Santa Clara, CA). GCaMP was excited at 900 or 920 nm and emission signals were collected by GaAsP photomultiplier tubes (PMTs). Frame images ($256 \times 256$ pixels) were acquired at 5–10 Hz. The region of interest (ROI) covers the entire bilateral medial γ5 lobe in MB. For consistency, imaging focus was kept approximately at the same level in different animal guided by axon position of M6 or aSP13.

## Optogenetic stimulation and functional connectivity

For SFOCatCh experiments, neurons were activated with whole field light from a mercury lamp (X-cite 120 PC, Excelitas Technologies). Light was filtered by 38HE 470/40 nm and 43HE 550/25 nm (Zeiss), and pulse duration was controlled by a TTL-triggered shutter (Uniblitz, Rochester, NY). Light density was calculated by dividing light power to fields of view (FOV) of objective: 480 nm, 0.28 mW/mm$^2$, and 540 nm, 0.86 mW/mm$^2$.

For CsChrimson experiments, LED (pE-4000, CoolLED) was used to deliver 2 ms light pulse as stimulation. Light (peak 635 nm) trains were further filtered by 635/18 nm (Semrock, Rochester, NY) and delivered at 30 Hz for 1 s. Light density was calculated as 0.126 mW/mm$^2$ when staying persistent during measurement at 635 nm.

## Focal dopamine perfusion

Dopamine (DA) solution was prepared freshly before each experiment. DA solution was back-filled into a glass electrode with fine tip (~3 µm) shortly before each focal application. DA was injected (1 s, 5 p.s.i) in the medial γ5 lobe by Picospritzer-III (Parker, Cleveland, OH) (*Cassenaer and Laurent, 2012*). We controlled for dopamine diffusion by co-loading a fluorescent dye (Texas red 3000, 0,1 mg/ml) to the focal pipette and monitoring the dye distribution area during two-photon scanning.

## Data analysis

GCaMP imaging data was analyzed in a custom program modified from *Sun et al., 2016*. Fluorescence changes in intensity were calculated as ΔF/F, where F is the average signals of the five frames before each stimulation. ROIs were chosen contained single optical plate with the neural processes of interest. Peak ΔF/F represents mean ΔF/F in continuous five frames responses (SFOCatCh ON) acquired during LTP procedure. All data represented as mean ± s.e.m. All statistical analyses were performed in Graphpad Prism 7.0a. Data were analyzed by unpaired Student's t-test or one-way ANOVA test with post hoc Tukey's range tests.

## Electrophysiology

For ex vivo patch-clamp recording from projection neurons (GH146-Gal4 > UAS SFOCatCh + UAS - mCD8::GFP) in antennal lobe brain explants were prepared as described in Ca imaging section, and recordings were performed as previously described (*Gu and O'Dowd, 2006*). Electrodes (5–7 MΩ) were filled with saline solution containing 140 mM potassium aspartate, 10 mM HEPES, 1 mM KCl, 4 mM MgATP, 0.5 mM $Na_3GTP$, 1 mM EGTA, pH 7.3, and 265 mOsm. Cell-attached recording was performed in voltage-clamp mode with 0 mV holding potential. Whole-cell patch-clamp recording was performed in current-clamp mode with resting membrane potential around 55–65 mV. Signals were acquired by MultiClamp 700B amplifier, digitized at 10 kHz, and low-pass filtered at 5 kHz.

## Acknowledgements

We thank Stefanie Wandl and Kristin Henderson for technical assistance, Gudrun Ihrke for help with immunohistochemistry, Karel Svoboda, Glenn Turner, Vivek Jayaraman and Ulrike Heberlein for comments on the manuscript. This work was supported by Howard Hughes Medical Institute-Janelia Research Campus, Boehringer Ingelheim GmbH-IMP Vienna, and Austrian Science Fund (FWF P24499 to KK).

## Additional information

### Funding

| Funder | Grant reference number | Author |
| --- | --- | --- |
| Howard Hughes Medical Institute | | Barry Dickson<br>Krystyna Keleman |
| Boehringer Ingelheim | | Barry Dickson<br>Krystyna Keleman |
| Austrian Science Fund | FWF P24499 | Krystyna Keleman |

The funders had no role in study design, data collection and interpretation, or the decision to submit the work for publication.

### Author contributions

Xiaoliang Zhao, Formal analysis, Investigation, Methodology, Designed, Conducted and interpreted the imaging and electrophysiology experiments; Daniela Lenek, Investigation, Designed, conducted, and analyzed the behavioral data (Figure 1); Ugur Dag, Investigation, Designed, conducted, and interpreted behavioral data (Figure 4); Barry J Dickson, Conceived and designed the overall study, Supervised the project, Interpreted the data, Generated SFOCatCH, Wrote the manuscript; Krystyna Keleman, Conceived and designed the overall study, Supervised the project, Interpreted the data, Funding acquisition, Wrote the manuscript

### Author ORCIDs

Xiaoliang Zhao https://orcid.org/0000-0001-8787-076X
Barry J Dickson https://orcid.org/0000-0003-0715-892X
Krystyna Keleman https://orcid.org/0000-0003-2044-1981

### Decision letter and Author response

Decision letter https://doi.org/10.7554/eLife.31425.021
Author response https://doi.org/10.7554/eLife.31425.022

## Additional files

### Supplementary files

• Supplementary file 1. Table S1. Constitutive silencing of MBγ neurons impairs STM. Courtship indices of naïve ($CI^-$) and experienced ($CI^+$) males of the indicated genotypes according to *Figure 1A*, tested in single-pair assays with mated females as trainers and testers, shown as mean ± s.e.m. and median (italics) of $n$ males. $P$ values determined by permutation test for the null hypothesis that learning equals 0 ($H_0$: SI = 0) or for the null hypothesis that experimental and control males learn equally well ($H_0$: SI = $SI_c$). Table S2. Constitutive silencing of M6 neurons impairs STM. Courtship indices of naïve ($CI^-$) and experienced ($CI^+$) males of the indicated genotypes according to *Figure 1B*, tested in single-pair assays with mated females as trainers and testers, shown as mean ± s.e.m. and median (italics) of $n$ males. $P$ values determined by permutation test for the null hypothesis that learning equals 0 ($H_0$: SI = 0) or for the null hypothesis that experimental and control males learn equally well ($H_0$: SI = $SI_c$). Table S3. Activation of MBγ neurons is more potent in experienced than naïve males. Courtship indices at $20^0$C ($CI^{20}$) and $32^0$C ($CI^{32}$) of naïve (-) or experienced (+) males of the indicated genotypes according to *Figure 1C*, tested in single-pair assays with

pseudomated females, shown as mean ± s.e.m. and median (italics) of $n$ males. All males where trained at room temperature. $P$ values determined by permutation test for the null hypotheses that learning equals 0 ($H_0$: SI = 0), that experimental flies do not differ from the controls ($H_0$: SI = $SI_c$), and that courtship is equally suppressed in experienced and naïve males ($H_0$: $SI^+$=$SI^-$). Table S4. Activation of M6 neurons is equally potent in naïve and experienced males. Courtship indices at $20^0$C ($CI^{20}$) and $32^0$C ($CI^{32}$) of naïve (-) or experienced (+) males of the indicated genotypes according to *Figure 1D*, tested in single-pair assays with pseudomated females, shown as mean ± s.e.m. and median (italics) of $n$ males. All males where trained at room temperature. $P$ values determined by permutation test for the null hypotheses that learning equals 0 ($H_0$: SI = 0), that experimental flies do not differ from the controls ($H_0$: SI = $SI_c$), and that courtship is equally suppressed in experienced and naïve males ($H_0$: $SI^+$ = $SI^-$).
DOI: https://doi.org/10.7554/eLife.31425.013

• Supplementary file 2. Table S5. Acute silencing of M6 neurons impairs STM acquisition and retrieval. Courtship indices of naïve ($CI^-$) and experienced ($CI^+$) males of the indicated genotypes according to *Figure 4H*, tested in single-pair assays at the indicated temperature ($^0$C) during training (Train) or testing (Test), with mated females as trainers and testers, shown as mean ± s.e.m. and median (italics) of $n$ males. $P$ values determined by permutation test for the null hypothesis that learning equals 0 ($H_0$: SI = 0) or for the null hypothesis that experimental and either type of control males learn equally well ($H_0$: SI = $SI_c$).
DOI: https://doi.org/10.7554/eLife.31425.014

• Supplementary file 3. Fly genotypes. Specific fly genotypes used in all main and supplementary figures.
DOI: https://doi.org/10.7554/eLife.31425.015

• Transparent reporting form
DOI: https://doi.org/10.7554/eLife.31425.016

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
