## [Decision Letter]

Thank you for submitting your article "Persistent activity in a recurrent circuit underlies courtship memory in *Drosophila*" for consideration by *eLife*. Your article has been reviewed by three peer reviewers, and the evaluation has been overseen by a Reviewing Editor and a Senior Editor. The following individuals involved in review of your submission have agreed to reveal their identity: Josh Dubnau (Reviewer #2).

The reviewers have discussed the reviews with one another and the Reviewing Editor has drafted this decision to help you prepare a revised submission.

Summary:

This impressive study by Zhao et al. addresses physiological correlates of courtship memories in *Drosophila*. The authors make use of the advantages *Drosophila* has to offer for addressing neural circuitry – genetic control over the activity of individual components of the local network to be studied – and combine these with pharmacological approaches. This study adds important knowledge on the physiological principles of the KC – MBON – DAN loop and also expands our understanding of the MB as central circuit during courtship conditioning.

The authors identify a novel role of the MBON M6 in memory acquisition and show that it works in concert with DANs that are activated for extended time periods via feed-forward excitation. Importantly the authors show that repetitive γ KC stimulation results in long lasting potentiation of the γ KC to M6 synapse, potentially reflecting the physiological correlate of short-term memory. This finding provides a possible link for recurrent activity and connects this to short-term memory formation

Essential revisions:

1) Subsection “Repetitive stimulation of MBg potentiates MBg to M6 transmission”. The authors describe "This suggests that, upon repetitive stimulation of MBg neurons, endogenous dopamine enables synaptic transmission to M6 neurons, obviating the need for any exogenous supply." to explain the long-lasting responses of M6 neurons in Figure 3. This can be tested in a straightforward manner by examining the activity of aSP13 neurons (the endogenous supplier of dopamine) following the same protocol.

2) If frequent (1 min interval) repetitive activation of MBg can induce prolonged responses of M6, can multiple co-application of MBg-activating light and DA every 3 min (modification of a protocol described in Figure 2) equally induce such responses as well? This protocol better resembles the stimuli that flies receive during courtship conditioning.

3) One concern is the potential discrepancy between the in vivo settings and the in vitro protocols used here. The authors should explicitly discuss potential differences between in vivo and in vitro settings, as the timing, strength and duration of sensory cues may differ significantly. An experiment that the authors may wish to consider to partially address this gap would be to test whether (a) their OFF/ON/OFF protocols elicit immediate responses (first trial) in M6 of brains of experienced (trained) males and if so (b) whether responses would decay with a similar time course to the artificial stimulation?

4) The findings presented strongly support a model in which MBgamma neurons are activated during training, leading to a recurrent/persistent feedback through M6 onto aSP13 DANs onto γ neurons. The authors state (Discussion section) that during training, "MBgamma neurons are likely to be frequently activated during the period in which aSP13 activity persists, leading to a longer lasting potentiation of MBgamma>M6 transmission and its behavioral correlate STM". And further, the authors conclude that the findings suggest a model in which "working memory, STM and LTM all reside in the same γ compartment", which is "in contrast to the prevailing view of memory progression in the *Drosophila* MB" where "distinct memory phases are located in different compartments or lobes".

But then how do the authors reconcile this with the lack of impact on courtship conditioning of shibireTS silencing of γ neurons (e.g. Baker lab, 2016) and the demonstrable impact of silencing in α/β and prime lobes? Something is amiss. To be clear, I'm not demanding a resolution of this. I think the same perplexity exists in the Pavlovian olfactory memory field where we can rescue DopR with only γ expression but blocking output from γ seems to have little impact. Rather, I'm arguing that this can’t be brushed under the carpet. It deserves explicit verbiage.

5) Throughout the manuscript, the authors make use of dual control via Gal4 and LexA, often switching which downstream target is under Gal4 or LexA control. The Gal4 or LexA line in question is usually labeled in the figures, but nowhere in the ms are the genetics adequately described. This should be added, e.g. in figure legends, and/or text, and/or Materials and methods section.

---

## [Author Response]

Essential revisions:1) Subsection “Repetitive stimulation of MBg potentiates MBg to M6 transmission”. The authors describe "This suggests that, upon repetitive stimulation of MBg neurons, endogenous dopamine enables synaptic transmission to M6 neurons, obviating the need for any exogenous supply." to explain the long-lasting responses of M6 neurons in Figure 3. This can be tested in a straightforward manner by examining the activity of aSP13 neurons (the endogenous supplier of dopamine) following the same protocol.

We have examined activity of aSP13 upon repetitive activation of the MBg neurons and indeed observed a gradual enhancement of calcium levels upon each successive stimulation (Figure 3). This result supports our conclusions that during repetitive activation of the MBgamma neurons, aSP13 provides the endogenous dopamine to facilitate transmission from MBg to M6 neurons (subsection “Repetitive stimulation of MBγ potentiates MBγ to M6 transmission”)

2) If frequent (1 min interval) repetitive activation of MBg can induce prolonged responses of M6, can multiple co-application of MBg-activating light and DA every 3 min (modification of a protocol described in Figure 2) equally induce such responses as well? This protocol better resembles the stimuli that flies receive during courtship conditioning.

When we repeatedly activated MBg at 3 min intervals we observed an enhanced response in M6 within the first few trials without the exogenous dopamine (see Author response image 1). This response grew slower than with 1 min intervals but also plateaued after ~20 repetitions, i.e. in this case, around 60 min. Thus, repetitive activation, even with longer intervals, is sufficient to induce enhanced response in M6 without addition of the exogenous DA (Author response image 1). We did not include this data in the manuscript as it does not bring any new insight into the proposed model of the recurrent circuit we describe in this study. With exogenous dopamine, M6 responds on the first trial (Figure 2).

**Author response image 1. respfig1:** Repetitive stimulation of MBg every 3 minutes potentiates MBg to M6 transmission without the exogenous DA. Time course of average DF/F responses in M6 dendrites during potentiation, mean+/-s.e.m.

3) One concern is the potential discrepancy between the in vivo settings and the in vitro protocols used here. The authors should explicitly discuss potential differences between in vivo and in vitro settings, as the timing, strength and duration of sensory cues may differ significantly. An experiment that the authors may wish to consider to partially address this gap would be to test whether (a) their OFF/ON/OFF protocols elicit immediate responses (first trial) in M6 of brains of experienced (trained) males and if so (b) whether responses would decay with a similar time course to the artificial stimulation?We agree that a prediction of our model, and the best way to directly examine the physiological consequences of behavioral training, would be to measure the first-trial responses of M6 neurons to MBg activation in trained males. Unfortunately, we have not had convincing results – either positive or negative – from our attempts to perform such experiments. Using explanted brains, we have not seen any clear differences in the first-trial responses of experienced and naïve males. We suspect that this is due to dopamine washout during the preparation of these samples. Accordingly, we attempted to monitor calcium responses using in vivo preparations by making an opening in the head cuticle. In males imaged shortly after a 1 hr training period, the GCaMP6s signal in M6 is notably elevated compared to naïve males already in the pre-stimulus OFF period (see Author response image 2). This is consistent with our model, but unfortunately makes it difficult to then reliably detect any further increase in the signal during the SFOCatCh ON period. In future, we hope to more tightly link the behavioral and physiological observations, first by electrophysiological recordings from M6 neurons of naïve and trained males, and second, by establishing a preparation in which we can train tethered flies so that these recordings can be performed longitudinally. Such experiments are currently (just) beyond our capabilities, and we believe the scope of the present study.

**Author response image 2. respfig2:** 

4) The findings presented strongly support a model in which MBgamma neurons are activated during training, leading to a recurrent/persistent feedback through M6 onto aSP13 DANs onto γ neurons. The authors state (Discussion section) that during training, "MBgamma neurons are likely to be frequently activated during the period in which aSP13 activity persists, leading to a longer lasting potentiation of MBgamma>M6 transmission and its behavioral correlate STM". And further, the authors conclude that the findings suggest a model in which "working memory, STM and LTM all reside in the same γ compartment", which is "in contrast to the prevailing view of memory progression in the Drosophila MB" where "distinct memory phases are located in different compartments or lobes".But then how do the authors reconcile this with the lack of impact on courtship conditioning of shibireTS silencing of γ neurons (e.g. Baker lab, 2016) and the demonstrable impact of silencing in α/β and prime lobes? Something is amiss. To be clear, I'm not demanding a resolution of this. I think the same perplexity exists in the Pavlovian olfactory memory field where we can rescue DopR with only γ expression but blocking output from γ seems to have little impact. Rather, I'm arguing that this can’t be brushed under the carpet. It deserves explicit verbiage.

First, we note that in the Baker, 2016 paper, one of the lines with expression in the MB γ dorsal neurons (MB419B) showed a strong courtship memory phenotype upon silencing with shibireTS (in the secondary screen that is actually identical to the primary screen). So it is not clear why the authors concluded that silencing of the γ neurons has no effect on courtship memory. In our hands, the MB γ dorsal and main lines which tested negative in the Baker, 2016 paper were consistently positive. We cannot explain why these lines were negative in their hands, whereas this other MB γ dorsal line was positive.

Secord, we independently tested the lines for α/β and α’/β’ neurons and confirmed their observation of a significant impairment of courtship suppression upon silencing (Author response image 3). Our results do not preclude involvement of other MB neurons/compartments in courtship memory, which we now explicitly note in the text (Discussion section).

**Author response image 3. respfig3:** KCs are required for courtship learning. Suppression indices (SI) in male flies in which active (*UAS-TNT*) or inactive (*UAS-TNT*Q) tetanus toxin was expressed in MB neurons. Statistical significance of differences from zero or from control groups is indicated as follows: *** *P*<0.001, *** P*<0.01*, *P<*0.05.

5) Throughout the manuscript, the authors make use of dual control via Gal4 and LexA, often switching which downstream target is under Gal4 or LexA control. The Gal4 or LexA line in question is usually labeled in the figures, but nowhere in the ms are the genetics adequately described. This should be added, e.g. in figure legends, and/or text, and/or Materials and methods section.

The newly added Supplementary file 3 lists full genotypes of flies used in all main and supplementary figures.